# Cadherin-11 contributes to the heterogenous and dynamic Wnt-Wnt-β-catenin pathway activation in Ewing sarcoma

Ryota Shirai[1], Tyler Biebighauser[1], Deandra Walker[1], Jillian Oviedo[1], Sarah Nelson-Taylor[1], Avery Bodlak[1], Timothy Porfilio[1], Naoki Oike[1,2], Andrew Goodspeed[3,4], Masanori Hayashi[1,3]*

1 Department of Pediatrics, University of Colorado School of Medicine, Aurora, Colorado, United States of America, 2 Division of Orthopaedic Surgery, Graduate School of Medical and Dental Sciences, Niigata University, Niigata, Japan, 3 University of Colorado Cancer Center, University of Colorado Anschutz Medical Campus, Aurora, Colorado, United States of America, 4 Department of Biomedical Informatics, University of Colorado Anschutz Medical Campus, Aurora, Colorado, United States of America

* masanori.hayashi@cuanschutz.edu

## Abstract

Ewing sarcoma is the second most common bone cancer in children, and while patients who present with metastatic disease at the time of diagnosis have a dismal prognosis. Ewing sarcoma tumors are driven by the fusion gene EWS/Fli1, and while these tumors are genetically homogenous, the transcriptional heterogeneity can lead to a variety of cellular processes including metastasis. In this study, we demonstrate that in Ewing sarcoma cells, the canonical Wnt/β-Catenin signaling pathway is heterogeneously activated in vitro and in vivo, correlating with hypoxia and EWS/Fli1 activity. Ewing sarcoma cells predominantly express β-Catenin on the cell membrane bound to CDH11, which can respond to exogenous Wnt ligands leading to the immediate activation of Wnt/β-Catenin signaling within a tumor. Knockdown of CDH11 leads to delayed and decreased response to exogenous Wnt ligand stimulation, and ultimately decreased metastatic propensity. Our findings strongly indicate that CDH11 is a key component of regulating Wnt//β-Catenin signaling heterogeneity within Ewing sarcoma tumors, and is a promising molecular target to alter Wnt//β-Catenin signaling in Ewing sarcoma patients.

## Introduction

While the intensification of conventional multi-modal therapy has significantly improved the survival rate for localized Ewing sarcoma from the less than 10% survival of the surgery-only era to the current approximate 70% survival, many patients still die from metastatic disease [1–3]. With intensive multi-modal therapy including chemotherapy, radiation, and surgery, many patients, even patients with initial metastatic disease, achieve a radiographic remission, although metastatic relapse after a remission is not uncommon. Furthermore, an initial good response to conventional therapy does not guarantee a cure. Ewing sarcoma carries few

**Data Availability Statement:** All relevant data are within the manuscript and its Supporting Information files. For genomic sequencing data,

this has been uploaded into GEO, and will be publicly available at GSE 263651.

**Funding:** The following includes all funding received to support this study. National Institutes of Health grant (K12HD068372) (M.H.), St. Baldrick's Foundation Scholar Award (M.H.), the Hyundai Hope on Wheels Young Investigator Award (M.H.), and Tanabe-Bobrow Foundation award (M.H.). This study was also partly supported by the National Institutes of Health P30CA046934 to the University of Cancer Center Bioinformatics and Biostatistics Shared Resource. All funders had no role in study design, data collection and analysis, decision to publish, or preparation of the manuscript. There was no additional external funding received for this study.

**Competing interests:** The authors have declared that no competing interests exist.

mutations other than the pathognomonic EWS/Fli1 gene fusion [4–6], and previous studies have demonstrated that while these tumors do not have genomic heterogeneity, there is transcriptional heterogeneity observed in cell lines and xenografts [7, 8]. The EWS/Fli1 fusion protein is an aberrant transcription factor that is essential to the oncogenesis of Ewing sarcoma, and multiple studies have led to the identification of over 1,000 genes that are directly or indirectly induced or repressed [9–12]. While this has led to a large body of work focused on identifying EWS/Fli1-induced genes that can be targeted to inhibit tumor proliferation, clinical translation has been difficult [13, 14].

While EWS/Fli1 expression is inarguably essential to the survival of Ewing sarcoma cells, emerging experimental and clinical evidence suggests that these tumors have significant intratumoral transcriptional heterogeneity, characterized by the heterogenous activation of EWS/Fli1 [8]. Conditional knockdown systems of EWS/Fli1 have indicated that low-level EWS/Fli1 promotes a pro-metastasis phenotype, compared to high expression [7]. However, EWS/Fli1 has been demonstrated to regulate a variety of cellular processes, suggesting a non-linear model of transcription regulation by EWS/Fli1 activity. Recent work in single cell transcriptomics further support this model, where EWS/Fli1 activation and gene regulatory signatures can be heterogenous from cell to cell [8]. Furthermore, EWS/Fli1 transcriptional regulation can also be regulated by cell extrinsic mechanisms such as the tumor microenvironment [15].

The Wnt signaling pathway is one of the key regulators of adult stem cell function, especially self-renewal and proliferation, as well as being involved in the pathogenesis of several malignancies [16–19]. This pathway is extremely complicated, involving 19 Wnt ligands and 10 Frizzled receptors which transduce signals via both canonical and non-canonical pathways [16]. In Ewing sarcoma, high Wnt signaling activity has been shown to be a predictor of poor clinical outcome [20]. While Ewing sarcoma tumors often do not have β-catenin activating mutations, and β-catenin nuclear accumulation is rarely found in clinical samples [21–23]. Several groups have reported the expression of different Wnt ligands and Frizzled receptors in Ewing sarcoma cell lines, and have demonstrated that morphologic changes or enhanced motility and invasion can be induced by exogenous Wnt ligands, indicating heterogenous activation of Wnt/ β-catenin signaling as a mechanism of metastasis progression [19, 24–27] In our previous work, we demonstrated that the Porcn inhibitor WNT974 can act as a pan-Wnt inhibitor, and suppress expression of a panel of genes related to epithelial-mesenchymal transition (EMT), resulting in phenotypic changes characterized by decreased motility and invasion [28]. In a mouse xenograft model, there was no significant effect on primary tumor growth, but we observed a striking delay in the development of hematogenous metastases which translated into prolonged survival. Here, we further report our investigation into Wnt signaling heterogeneity within Ewing sarcoma tumors, which are potentially regulated by the expression of Cadherin-11.

## Results

### β-catenin depletion leads to reduced metastases in Ewing sarcoma cells

Previously, our group demonstrated that inhibiting Wnt ligand secretion by Porcn inhibition delays metastasis formation in Ewing sarcoma [28]. Furthermore, it has been reported that the activation of canonical Wnt/β-Catenin signaling by exogenous ligands lead to decreased EWS/Fli1 expression, and increased migratory changes. Here, to establish that canonical β-Catenin signaling is in fact the key modulator of the metastatic phenotype observed in Ewing sarcoma, we established β-Catenin knockdown cells. β-Catenin was knocked down in Ewing sarcoma cell lines TC71 and A4573 using lentivirally delivered shRNAs (Fig 1A). Knockdown of β-Catenin led to significant decrease in colony formation, measured by clonogenic assays

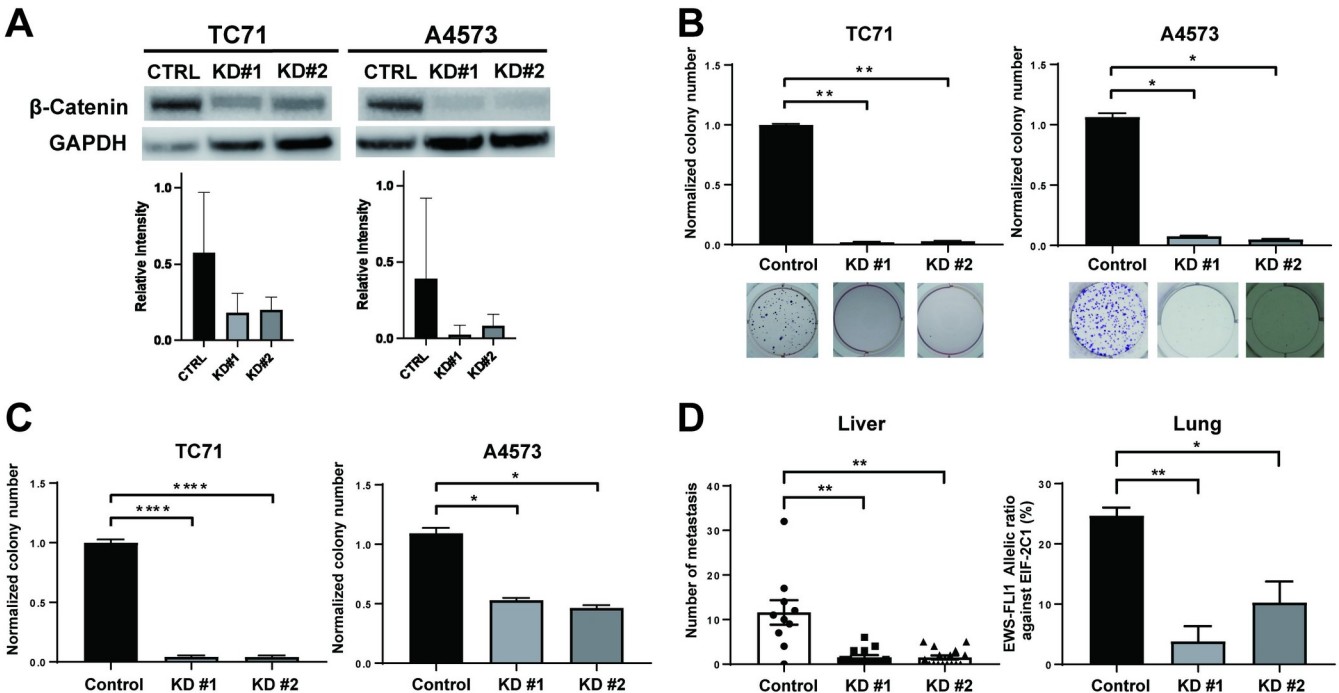

**Fig 1. β-catenin depletion leads to reduced clonogenicity, anchorage independent growth, and metastatic dissemination in Ewing sarcoma cells.** (A) β-catenin was stably knocked down by shRNA (KD#1 and KD#2), and was confirmed by immunoblotting. Control is shRNA targeting GFP (CTRL). (B) Effect of β-catenin shRNA-knockdown significantly decreased colony formation in clonogenic assays. Shown are quantification of colonies per wells, as well as representative wells. (C) In non-adherent soft agar colony formation assays, β-catenin knockdown led to a significant decrease in colony formation. (D) in vivo tail vein injection models demonstrate that β-catenin knockdown leads to less metastatic dissemination to the liver as well as lungs. Quantifications represent mean and standard error of the mean from three or more experiments performed independently. Statistical significance was determined using a Mann-Whitney Test with multiple comparisons. ns, not significant, *p<0.05, **p<0.01, ****p<0.0001.

(Fig 1B), as well as a significant decrease in colony formation in soft agar assays (Fig 1C), indicating inhibition of anchorage dependent and independent growth. β-Catenin knockdown did not lead to significant change of cell proliferation in regular cell culture conditions (S1 Fig in S1 File). Next, to test whether β-Catenin contributes to the stable engraftment of metastatic dissemination, control cells and β-Catenin knockdown cells were injected via the tail vein. Mice were followed for 4 weeks, and liver and lung tissue were harvested. Here, compared to scramble shRNA control cells, β-Catenin knockdown cells produced a significant reduction in metastatic disease burden in both liver and lungs (Fig 1D).

## Canonical Wnt/β-Catenin pathway activation is heterogenous in vivo and in vitro

Previously, Ewing sarcoma tumors have been shown to have rare cells with nuclear or cytoplasmic β-Catenin expression, and an ability to respond to exogenous Wnt ligands *in vitro* [20]. However, whether Wnt/β-Catenin signaling is heterogeneously activated without exogenous Wnt ligands remains unclear. Here, we used the 7TGC fluorescent reporter to assess for canonical β-Catenin dependent transcription. The 7TGC reporter has a constitutively active SV49-mCherry cassette as well as a 7xTcf-eGFP reporter cassette, which can be used to assess for Wnt/β-Catenin signaling activation through eGFP expression [29]. Three Ewing sarcoma cell lines TC71, A4573, and SK-ES1 were stably transduced with the 7TGC reporter. When these cells in adherent culture conditions were assessed, only 0.012–0.06% of cultured cells

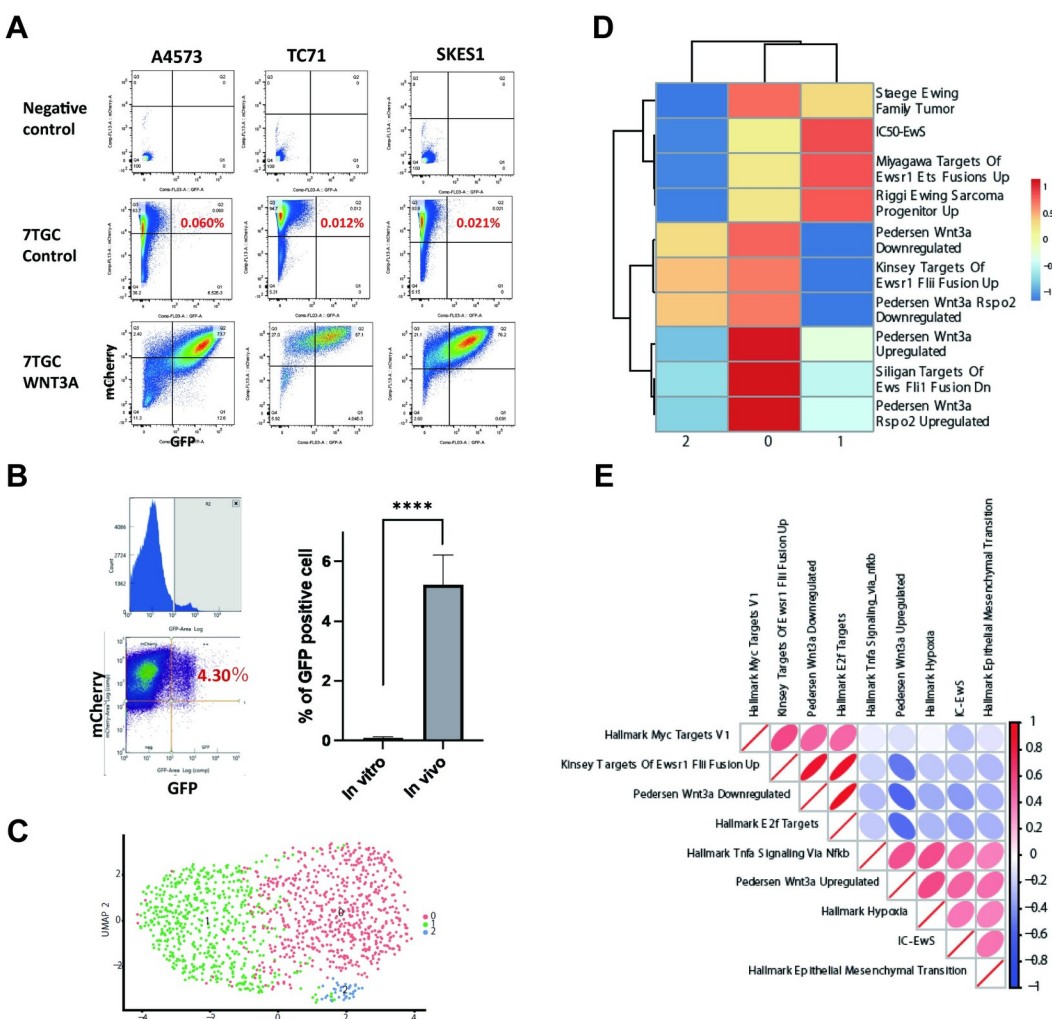

**Fig 2. Wnt/β-catenin pathway activation is heterogenous within Ewing sarcoma tumors.** (A) Flowcytometry analysis of cultured Ewing sarcoma cell lines stably expressing the Wnt reporter 7TGC, demonstrate a small population of mCherry and eGFP expressing cells. (B) 7TGC expressing A4573 xenografts demonstrate a small subpopulation of mCherry and eGFP positive cells *in vivo.* The proportion of eGFP positive cells are significantly higher *in vivo* than in vitro. (C, D) Single cell RNA-seq of a A4573 xenograft demonstrates significant heterogeneity of Wnt pathway activation. (E) Tumor cells within A4573 xenografts demonstrate positive and negative correlation of various pathways with the IC-EwS signature. Fisher exact test was used to compare proportion of mCherry and eGFP positive cells in Fig 2B. ****p<0.0001.

expressed eGFP, indicating a very small portion of cells are highly active in Wnt/β-Catenin dependent transcription, consistent with previous reports using other cell lines [20] (Fig 2A). Next, A4573 cells were implanted orthotopically in the pretibial space of mice to assess for *in vivo* activation as previously described [30]. Tumors were harvested after 2 weeks of growth, dissociated to single cells suspensions, then assessed by flowcytometry for mCherry and eGFP expression. Here, we found that in the *in vivo* tumor microenvironment, there is a significant increase of cells positive for eGFP, demonstrating a subpopulation of cells with highly active Wnt/β-Catenin dependent transcription (Fig 2B). Next, we further performed single cell RNA-seq on these xenografts to assess for downstream changes induced by this Wnt/β-Catenin activation heterogeneity, and whether this correlated with any EWS/Fli1 signature. Here, we identified three clusters, and cluster 0 demonstrated a significant enrichment for the previously

published Pedersen Wnt3 upregulated activity gene set [20], as well as for hallmarks of Hypoxia, and Hallmarks of EMT, consistent with previous *in vitro* findings (Fig 2C and 2D). Cluster 2 was enriched for the recently published single cell RNAseq based EWS/Fli1 signature by Aynaud et al (IC-EwS) [8]. To further assess for correlation between specific gene sets across all cells, we further analyzed the single cell RNAseq data for correlation across gene sets. Here, we found that Wnt3a responsive upregulated gene scores significantly correlate for a positive enrichment for hypoxia, EMT, while also negatively enriching for the Kinsey targets of EWS/Fli1 fusion up regulation, as previously reported (Fig 2E). When analyzing for correlation with IC-EwS, we also were able to find a strong correlation with the Wnt3a response gene signature, as well as for hallmarks of TNFα, EMT, and hypoxia, while proliferative pathways such as hallmarks of G2M checkpoint, E2F targets negatively correlate. While this was surprising, perhaps this indicates that the highly Wnt responsive cells within the in vivo environment are in hypoxic environments, undergoing EMT like changes, and have high activity of EWS/Fli1 responsive genes.

## β-Catenin localizes in the cell membrane and cytoplasm of Ewing sarcoma cells

Despite having only a subpopulation with evidence of robust Wnt/β-Catenin signaling, Ewing sarcoma has been reported to be a tumor group with high β-Catenin expression [31]. To further investigate the expression of β-Catenin in Ewing sarcoma cells, we next performed immunostaining for β-Catenin, where we found that very few cells express nuclear β-Catenin, but almost all cells express an abundance of β-Catenin in the cytoplasm and cell membrane (Fig 3A and 3B). Furthermore, western blots on fractionated cellular proteins from cultured cells demonstrate that β-Catenin is most commonly localized in the membranous fraction,

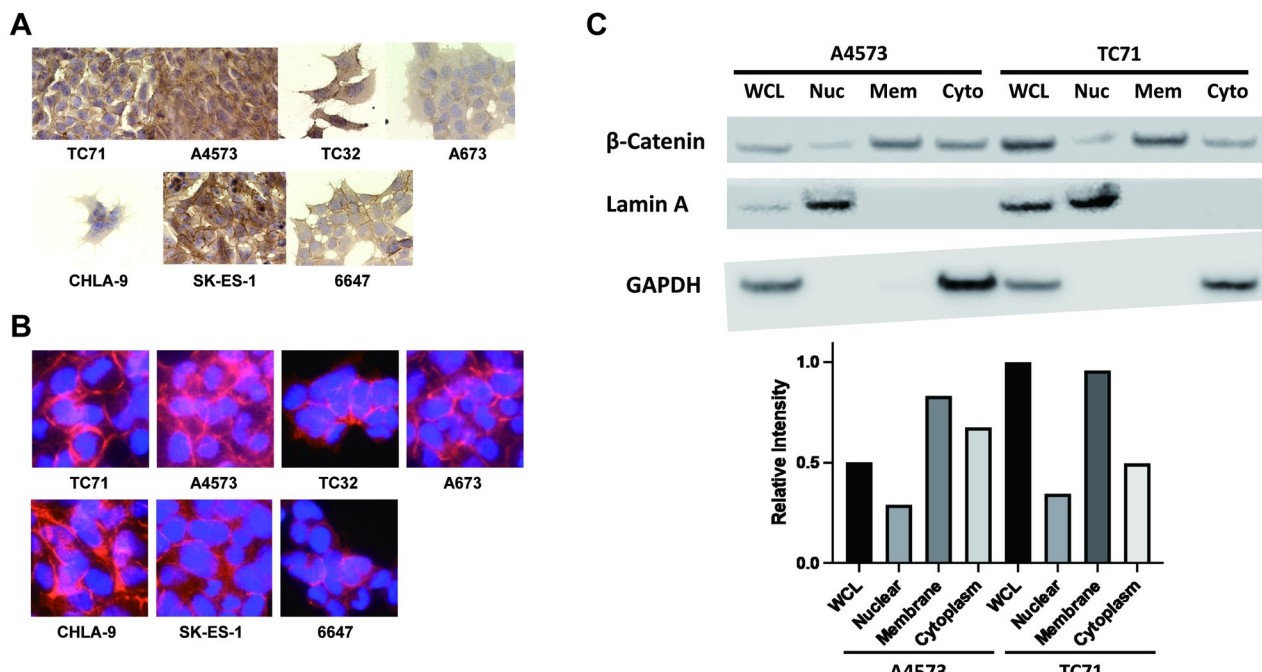

**Fig 3. β-catenin localizes in the cell membrane and cytoplasm in Ewing sarcoma cells.** (A) Immunohistochemistry confirms that β-catenin is highly localized in the cellular membrane. (B) Immunofluorescent staining demonstrates that β-catenin is highly expressed in the cytoplasm of Ewing sarcoma cells. (C) Western blot analysis of cellular fractions of ES cells demonstrate that β-catenin is abundantly expressed in the membranous fraction (Mem), and cytoplasmic fraction (Cyto), and minimally in the nuclear fraction (Nuc). Whole cell lysate (WCL) was loaded as control. Relative intensity of β-Catenin band measured by densitometry shown.

followed by the cytoplasmic fraction, with scant nuclear β-Catenin (Fig 3C). Overall, these results indicate that Ewing sarcoma tumors highly express β-Catenin in the cell membrane and cytoplasm in the majority of cells, while a small subpopulation of cells have robust β-Catenin driven transcription.

### Ewing sarcoma cells highly express CDH11, which is bound to β-Catenin on the Cell membrane

Addition to functioning as a transcription factor driving the canonical Wnt pathway, β-Catenin is well known to be an important component of the adherens junction on epithelial cells. Adherens Junctions are complexes that drive cell-cell adhesion, and composed of cadherin adhesion receptors. p120, α-, and β-catenin. E-Cadherin, and N-Cadherin are critical components of this adherens junction in many cancer cells. Considering the localization of β-catenin at the cell membrane of Ewing sarcoma cells, we next explored whether this localized β-catenin is a component of the adherens junction in Ewing sarcoma. To our surprise, we discovered that Ewing sarcoma cells do not express traditional adherens junction proteins E-Cadherin nor N-Cadherin, but highly express CDH11, a bone Cadherin (Fig 4A). On immunofluorescence, CDH11 highly co-localizes with β-catenin on the cell membrane (Fig 4B). Furthermore, upon co-immunoprecipitation, western blot demonstrates that CDH11 directly binds to β-catenin (Fig 4C). Overall, these findings indicate that the highly expressed β-Catenin in Ewing sarcoma directly binds and localizes with CDH11.

### β-catenin dependent Wnt-β-catenin signaling is dependent on CDH11 expression

Considering that CDH11 and β-catenin directly bind on the cell membrane, we next investigated whether there is a correlation between CDH11 and β-catenin expression. First, using

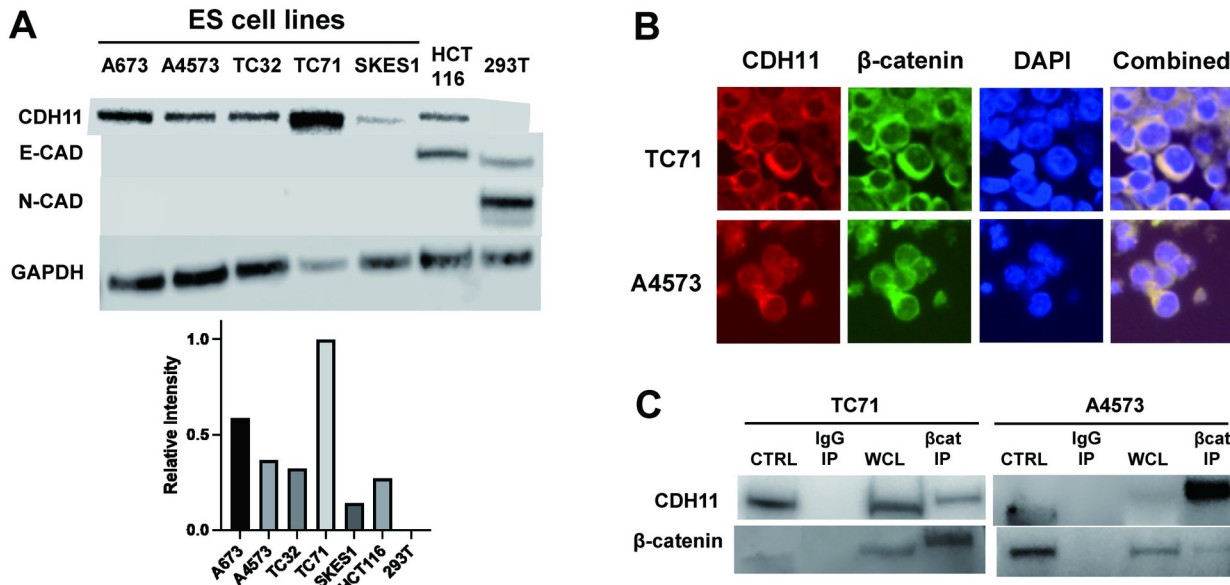

**Fig 4. CDH11 is highly expressed by Ewing Sarcoma cells, and is bound to β-catenin.** (A) Ewing sarcoma cells were assessed for Cadherin expression, demonstrating strong CDH11 expression, while E-Cadherin and N-Cadherin is not expressed. Relative intensity of CDH11 band measured by densitometry shown. (B) Immunofluorescence of Ewing sarcoma cell lines demonstrates that CDH11 and β-catenin co-localize on the cytoplasmic portion of TC71 and A4573 cells. (C) Co-Immunoprecipitation shows that CDH11 is bound to β-catenin in Ewing sarcoma cells, while IgG input control (IgG IP) does not detect β-catenin.

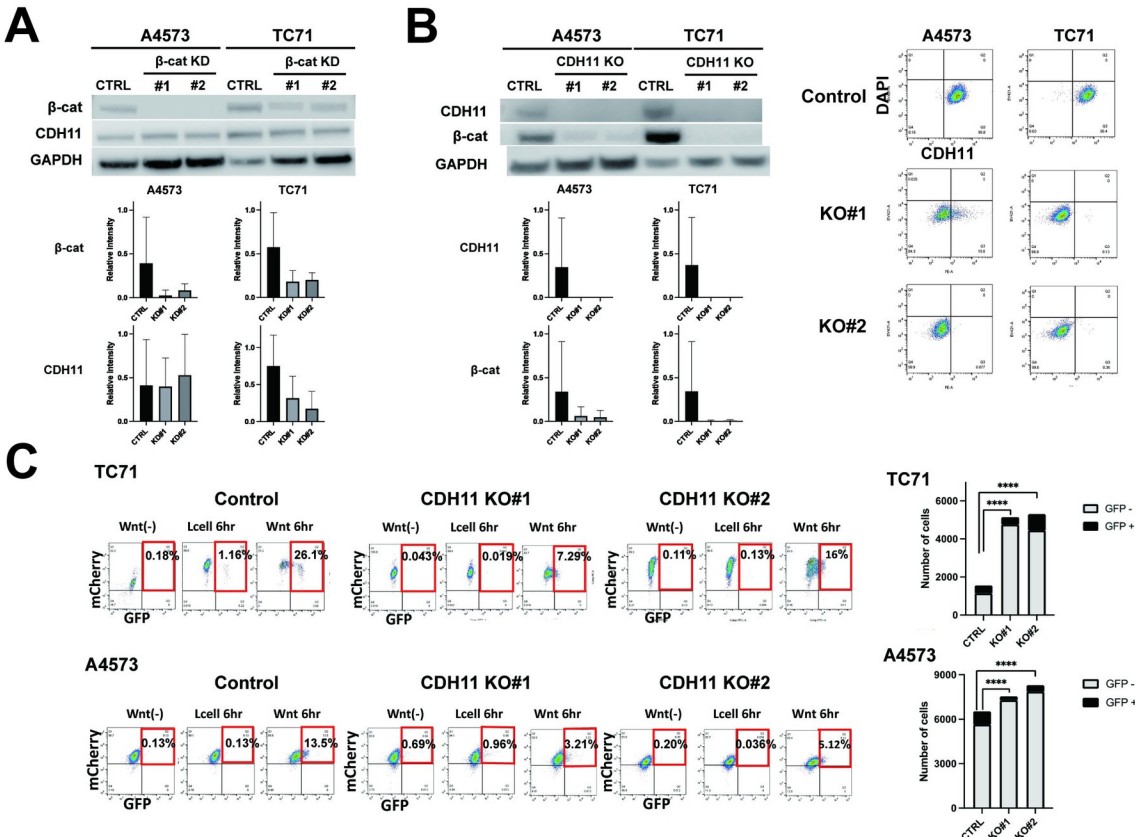

**Fig 5. Stable β-catenin expression, and Wnt ligand response is dependent on CDH11.** (A) β-catenin knockdown in Ewing sarcoma cells do not lead to alteration of CDH11 expression. (B) CDH11 knockout leads to loss of β-catenin expression. Flowcytometry confirms successful CDH11 knockout. Relative intensity of β-Catenin and CDH11 bands measured by densitometry shown. (C) 7TGC expressing Ewing sarcoma cells were exposed for six hours to supernatant from L cells expressing a Wnt3a overexpression vector (Wnt 6hr), wild type L cells as a control (L cell 6 hour), or regular culture media (Wnt(-)). Compared to control cells transfected with empty control vectors (Control), CDH11 knockdown cells have a delayed and diminished response to exogenous Wnt3a, measured by eGFP expression. Quantifications represent mean and standard error of the mean from three or more experiments performed independently. Statistical significance was determined using a Mann-Whitney Test with multiple comparisons. Fisher exact test was also used to compare proportion of mCherry and eGFP positive cells in Fig 5C. ns, not significant, ****p<0.0001.

shRNA knockdown, we found that β-catenin knockdown does not lead to a change in CDH11 expression (Fig 5A). However, CDH11 knockout led to a reduction in β-catenin expression (Fig 5B). Interestingly, when β-catenin mRNA was assessed through quantitative real-time PCR, expression was variable in CDH11 knockouts, indicating that CDH11 knockout is contributing to the altered stability of the β-catenin protein (S2 Fig in S1 File). Considering that β-catenin is the driver of canonical Wnt signaling, we further investigated whether CDH11 depletion can affect canonical Wnt signaling in Ewing sarcoma cells. Here, Ewing sarcoma cells were exposed to exogenous Wnt ligands, and Wnt-β-catenin signaling response was assessed by the 7TGC Fluorescent reporter. Here, control cells were found to express eGFP indicating Wnt/β-catenin pathway activation after 6 hours of Wnt3a exposure. However, CDH11 knockdown cells had a significant decrease in cells with eGFP expression following Wnt3a exposure (Fig 5C). These cells were further assessed for β-catenin localization with cell fractionation western blots. Here, while there appears to be an overall decrease in β-catenin expression in CDH11 knockout cells (as also seen in Fig 5B), the cell fractionation could not

detect significant differences in nuclear β-catenin localization, perhaps due to the poor sensitivity of this assay in detecting small subpopulations (S3 Fig in S1 File).

## CDH11 depletion inhibits Ewing sarcoma colony formation and migration

Based on our earlier findings that demonstrated β-catenin depletion leads to decreased colony formation, and metastasis in Ewing sarcoma cells, next, we tested whether CDH11 depletion can also lead to decreased metastatic propensity. Here, as predicted, CDH11 depletion did lead to decreased colony formation in a clonogenic assay (Fig 6A). However, in a soft agar assay, while there was a tendency towards decreased anchorage independent growth, we did not find a statistical difference after CDH11 depletion (Fig 6B). The proliferation rate was not affected by CDH11 depletion (S4 Fig in S1 File). Interestingly, we noticed a significant change in the morphology of CDH11 knockout cells, represented by a decrease of cell surface area, which was similar to our previous findings when ES cells were treated with the pan-Wnt inhibitor WNT974 [28]. (Fig 6C). Addition to anchorage independent growth, the cell migratory ability has been known to be metastasis promoting. When CDH11 depleted cells were assessed for chemotactic migration, CDH11 knockout cells were found to have significantly lower

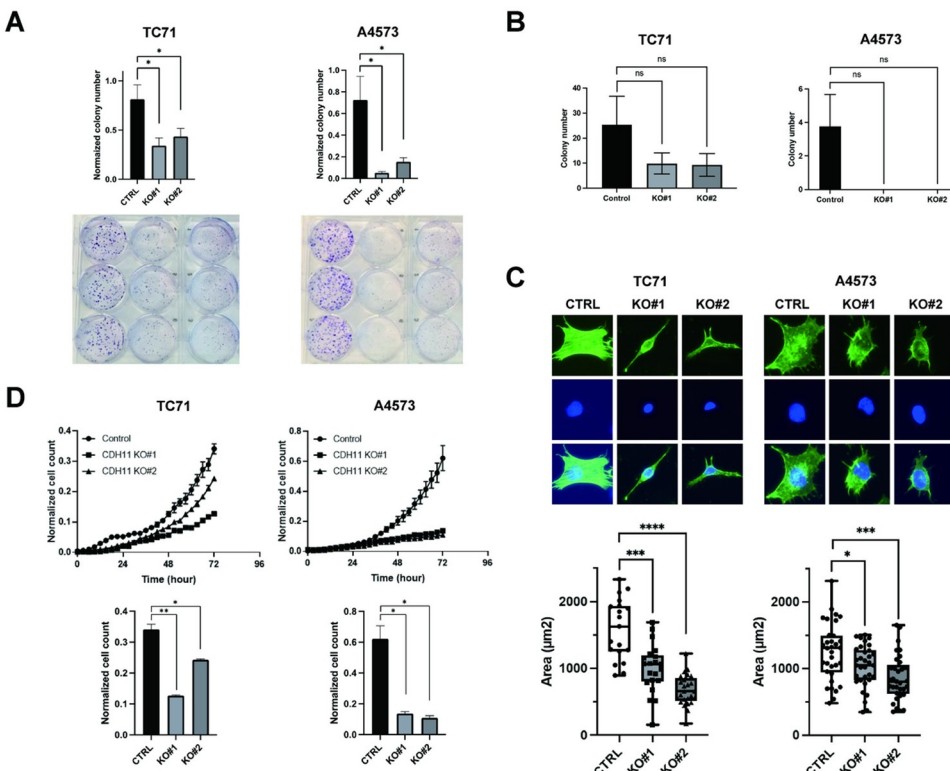

**Fig 6. CDH11 depletion in ES cells lead to reduced clonogenicity, migration, and morphology. (A)** CDH11 knock-out leads to significant reduction in colony number in clonogenic assays. **(B)** In anchorage independent soft agar growth assays, CDH11 knockdown leads to a trend towards a decrease in colony growth. **(C)** CDH11 knock-out leads to a significant reduction in the percentage of cells exhibiting neurite outgrowth, leading to a significant decrease of the area of cells. Representative images as well as measurement of cell area is demonstrated. **(D)** CDH11 knockout results in significant reduction of migration measured by chemotaxis migration in A4573 cells. Quantifications represent mean and standard error of the mean from three or more experiments performed independently. Statistical significance was determined using an unpaired T-test and Mann-Whitney Test with multiple comparisons. ns, not significant, *p<0.05, **p<0.01, ***p<0.001, ****p<0.0001.

migration compared to control in A4573 cells (Fig 6D). Taken together, our findings suggest that CDH11 depletion leads to a less migratory, less clonogenic cell phenotype.

## CDH11 depletion leads to decreased metastatic outgrowth of Ewing sarcoma cells

Finally, based on the above *in vitro* findings, we tested whether CDH11 depletion leads to decreased metastasis *in vivo*. Here, we injected CDH11 knockout cells, and cells transfected with empty controls through the tail vein of mice, and observed for formation of metastasis. In this model system, CDH11 knockout did in fact lead to a significant reduction of metastatic burden in the lungs in A4573 and one knockout clone in TC71, measured by the number of EWS-FLI1 copies measured by ddPCR in the lung tissue (Fig 7A). This model system can often produce liver metastases, a phenomenon not observed commonly in human patients, but nevertheless been studied in this model system. When liver metastasis numbers were enumerated visually, CDH11 knockout demonstrated variable results with a statistical decrease in one knockout clone for both TC71 and A4573, and no change in the other knockout clone (Fig 7B). Next, we tested whether this tendency towards decrease of metastasis would lead to extended survival, and tested the CDH11 knockout cells in our orthotopic-Implantation-Amputation model [30]. Here, we found that in the first knockdown, there was a significant improvement in survival, while the second knockdown had a tendency toward delayed death, however with no significance (Fig 7C). To investigate whether metastatic lesions had re-expressed CDH11, we assessed for continued CDH11 and β-catenin expression in the harvested tumors, and confirmed that even outgrowth metastatic lesions in the CDH11 knockdown group still did not express CDH11 nor β-catenin (Fig 7D).

## Discussion

In this study, we demonstrate that in Ewing sarcoma cells, β-Catenin expression is dependent on CDH11 expression, and the robust activation of the Wnt/β-Catenin pathway is partially reliant on CDH11. Loss of CDH11 can lead to a less metastatic phenotype, that is less responsive to Wnt ligands in the tumor microenvironment. In previous work, ES cells have been demonstrated to be highly responsive to exogenous Wnt ligands to induce a highly metastatic phenotype, which has also been shown to be targetable with Porcn inhibitors acting as pan-Wnt inhibitors [20, 28]. While we have recently published the efficacy of pan-Wnt inhibition through Porcn inhibitors in Ewing sarcoma, here, we sought to focus on the heterogeneity of Wnt/β-Catenin activation in the tumor, and alternative methods to target the responsiveness of Ewing sarcoma cells to exogenous ligands.

Cadherins have been known to be a key component of epithelial-mesenchymal transition, which contributes to carcinoma metastasis. E-Cadherin and N-Cadherin, the most dominant cadherins expressed in carcinomas significantly contribute to metastasis [32, 33], and has been known to suppress Wnt/β-Catenin pathway activation through sequestration of β-Catenin to the cell membrane [34, 35]. Furthermore, activation of Wnt/β-Catenin signaling can lead to downregulation of E-Cadherin, increase of N-Cadherin, and EMT [35]. In this work, we demonstrate that in ES cells, CDH11 is the dominant cadherin expressed, and the CDH11-β-catenin complex is highly expressed on the cell membrane and cytoplasm. The loss of this complex through loss of CDH11 can lead to the destabilization of β-catenin, and decreased response to Wnt ligands. This phenomenon has been previously been described in CML cells, where the N-Cadherin/β-catenin complex have been found to function in a similar fashion [36]. Similarly in melanoma cells, previous reports have suggested that during transendothelial migration, melanoma cells undergoes a dissociation of it's N-Cadherin-β-catenin complex, leading

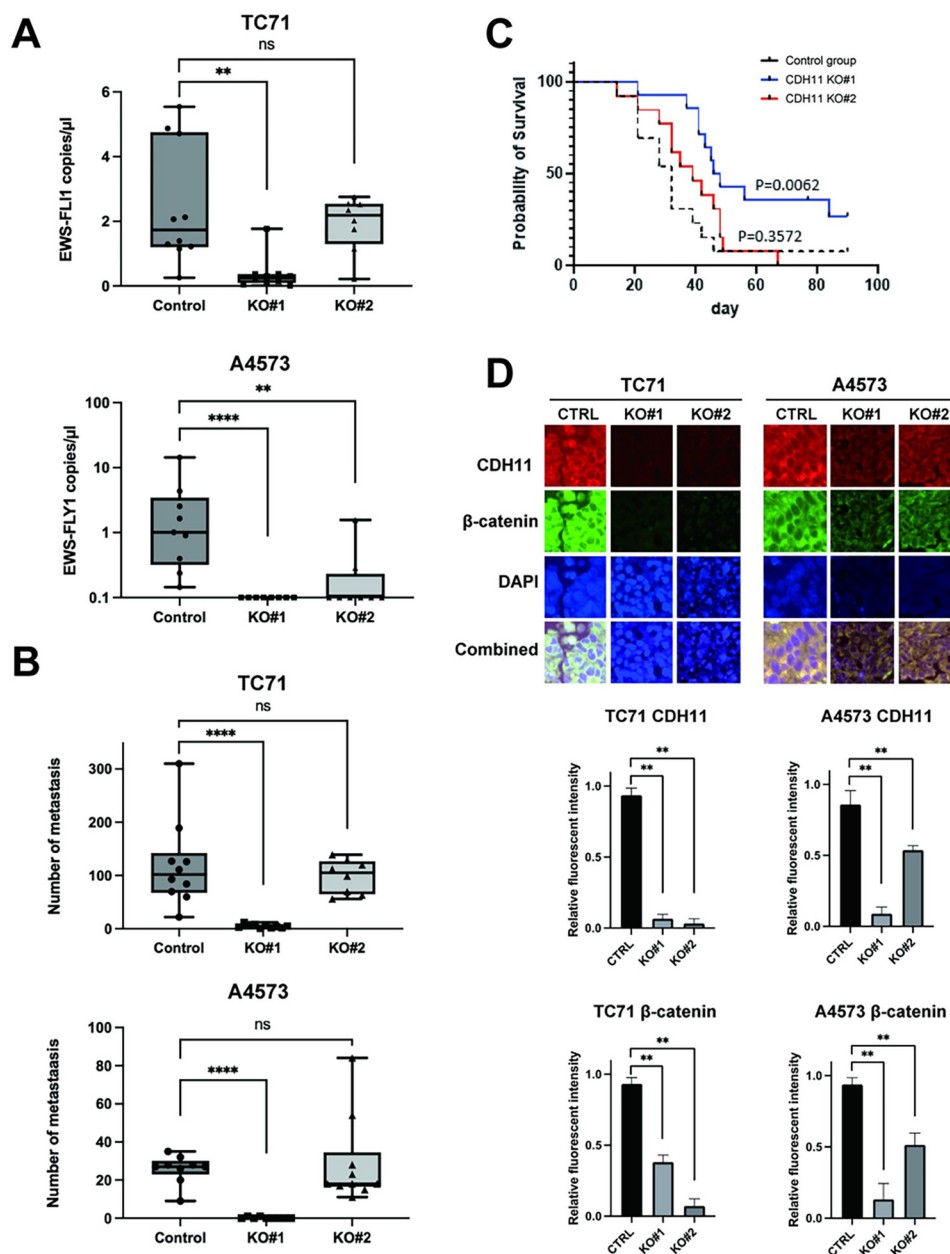

**Fig 7. CDH11 depletion leads to decreased Ewing sarcoma metastasis.** (A-B) CDH11 knockout cells develop less metastatic burden in the lung and liver, following tail vein injection. (C) Kaplan-Meier Survival curve of overall survival for control and CDH11 knockout cells. (D) immunofluorescence demonstrates sustained CDH11 knockout in outgrown tumors. Statistical significance was determined using a Mann-Whitney Test with multiple comparisons. ns, not significant, **p<0.01, ****p<0.0001.

to the immediate activation of β-catenin mediated gene transcription [37] and increased migration. Our results suggest that the CDH11-β-catenin complex may regulate β-catenin driven transcription through a similar fashion in Ewing sarcoma cells. In ES cells, the contribution of CDH11 to cell adhesion and metastasis has been previously reported [38]. Here, we demonstrate that CDH11 in Ewing sarcoma functions not only as an adhesion molecule, but also as a key regulator of Wnt ligand responsiveness, a key contributor to metastatic progression.

Cadherins have been increasingly recognized as potential anti-metastasis targets [32, 33], as well as for it's contribution to inflammation and fibrosis [39, 40]. While we have previously demonstrated that targeting Wnt signaling activation through WNT974 can lead to decreased metastasis in our OIA mouse model, targeting the complex Wnt signaling pathway clinically continues to be challenging, with mixed results in early phase trials [41]. Our data suggests that CDH11 is a key regulator of Wnt ligand response in ES cells, and can be a potential anti-metastasis therapeutic target for Ewing sarcoma patients.

## Materials and methods

### Cell lines

Ewing sarcoma cell lines TC71, A4573, A673, TC32, SK-ES-1 were kind gifts from Dr. David Loeb (Albert Einstein School of Medicine, Bronx, NY). Human colon cancer cell line HCT116 and human embryonic kidney cell line 293T were from the American Type Culture Collection (ATCC Manassas, VA). Cell line identity was confirmed by STR profiling in the Molecular Biology Core Facility of the University of Colorado School of Medicine (Aurora, CO). L-Cells and Wnt3a producing L-Cells were obtained from Dr. Melanie Koenigshoff (University of Colorado, Aurora CO). Cells were cultured at 50–70% confluence in Roswell Park Memorial Institute (RPMI) 1640 medium or Dulbecco's Modified Eagle Medium (DMEM) supplemented with 10% fetal bovine serum (FBS; Invitrogen, Grand Island, NY) and were routinely confirmed to be Mycoplasma negative using the MycoAlert Plus Mycoplasma detection kit (Lonza, Allendale, NJ).

### Quantification for Wnt/β-catenin signaling activity using flow cytometry

7xTcf-eGFP/SV40-mCherry reporter lentivirus (7xTGC) system was used for quantifying Wnt/β-catenin signaling activity. 293T cells were co-transfected with 7xTGC (#24304, Addgene) vector along with a 2nd generation packaging plasmid, psPAX2 and pMD2.G. Viral supernatants were harvested until 72 hours after the transfection and used to transduce Ewing sarcoma cells. Stably transduced cells for mCherry expression were selected by FACS by a MoFlo XDP100 (Beckman Coulter, Brea, CA). GFP expression was detected and quantified using BD FACSCelesta Cell Analyzer (BD biosciences, Franklin Lakes, NJ) 6, 24 hours after adding WNT3A conditioned L-Cell medium (WNT3A medium) or L-Cell medium and before adding any ligand as a control.

### shRNA knockdown

Lentiviral transduction was performed as previously described [42]. For β-catenin knockdown, cell lines were transduced with pLKO.1.-puro lentiviral short hairpin RNAs vectors targeting β-catenin expression (#19761and #19762, Addgene) using a 3rd generation packaging plasmid, pMDLg/pRRE (#12251, Addgene), pRSV-Rev (#12253, Addgene) and pMD2.G (#12259, Addgene). Control cells were transduced with pLKO.1 GFP shRNA (#30323, Addgene). Cells were selected in puromycin (2 μg/mL) before use in subsequent experiments. 7TGC was a gift from Roel Nusse (Addgene plasmid # 24304). pLKO.1.sh.beta-catenin.1248 and pLKO.1.sh.beta-catenin.2279 was a gift from William Hahn (Addgene plasmid # 19761 and # 19762). pMDLg/pRRE, pRSV-Rev, and pMD2.G was a gift from Didier Trono (Addgene plasmid # 12251, #12253, and #12259). pLKO.1 GFP shRNA was a gift from David Sabatini (Addgene plasmid # 30323).

### CRISPR/Cas9 knockouts

CRISPR/Cas9 knockouts of Ewing sarcoma cell lines were generated using the px458 / pSpCas9(BB)-2A-GFP (#48138, Addgene). pSpCas9(BB)-2A-GFP (PX458) was a gift from

Feng Zhang (Addgene plasmid # 48138). The CRISPR/Cas9 vectors were cloned at the University of Colorado Cancer Center Functional Genomics Core Facility, using the following F and R Oligos for each gRNA, which were annealed and ligated into BbsI digested px458 as previously described [43].

CDH11_1009_2_F `CACCGCGTGCCTGAGAGGTCCAATG`

CDH11_1009_2_R `aaacCATTGGACCTCTCAGGCACGc`CDH11_1009_3_F `CACCGCTTGGGGCCAAGAACATAGG`

CDH11_1009_3_R `aaacCCTATGTTCTTGGCCCCAAGc`

Targeted cells were transfected with Lipofectamine 2000 (Thermo Fisher Scientific, Waltham, MA) according to the manufacturer's instructions, then cells with high GFP expression were selected by FACS, and clones were selected for appropriate knockout of the target gene. Control cells were transfected with an empty px458 backbone.

### Clonogenic assay

One thousand cells were seeded into 9.6cm$^2$ plates with 2ml of RPMI-1640 medium. After 1 week, the number of colonies were enumerated after staining with crystal violet. The experiment was performed in triplicate.

### Colony formation assay in soft Agarose

$4 \times 10^3$ cells were mixed with 0.3% UltraPure Low Melting Point Agarose (Thermo Fisher Scientific, Waltham, MA) in RPMI-1640 media with 10% FBS and plated onto six-well plates containing a solidified bottom layer (0.6% UltraPure Low Melting Point Agarose in growth medium). Colonies were enumerated 21 days after preparation. The colonies larger than 0.1mm in diameter were scored as positive. The experiment was performed in triplicate.

### Proliferation assays

Cells were plated in 96 well plates in RPMI-1640 media with 10% FBS at a density of 5x10$^3$ cells per well. After 48 hours of incubation, cells were assessed for viability with the Cell Counting Kit-8 (Dojindo Molecular Technologies, Rockville, MD) per manufacturer's instructions. The experiment was performed in triplicate.

### Incucyte Chemotaxis Migration assays

Chemotaxis Migration assays were performed using IncuCyte Clearview 96-well plate for Chemotaxis (#4582, Sartorius AG, Göttingen, Germany). 5.0 x10$^3$ tumor cells (TC-71 and A4573) which were transduced with Incucyte Nuclight Red lentivirus EF1a Puro (Sartorius AG, Göttingen, Germany), were suspended in 60 μL of RPMI-1640 medium with no FBS in the upper chamber. 200 μL of RPMI-1640 medium with 20% FBS were placed as the chemoattractant in the lower chambers. Images of cells at the upper and bottom surfaces of each ClearView membrane were obtained every 3 hours for 72 hours using an IncuCyte live-cell Imaging System. The IncuCyte Analysis software (Sartorius AG) was used to quantify migration of tumor cells, normalized to the initial upper chamber seeded cell count.

### Cell fractionation

The cell fractionation was carried out using Cell Fractionation kit (#9038 Cell Signaling Technology, Danvers, MA) following the manufacturer's instructions.

## Co-immunoprecipitation

Immunoprecipitations were performed using the Pierce Co-immunoprecipitation Kit (Thermo Fisher scientific, Waltham, MA) according to the manufacturer's instructions.

## Western blotting

Harvested cells were rinsed in ice-cold PBS and lysed using Qproteome Mammalian Protein Prep kit (Qiagen, Valencia, CA). After centrifugation at 14000rpm for 10 minutes, protein concentrations were measured using Pierce BCA Protein Assay kit (Thermo Fisher scientific, Waltham, MA). Except for the western blotting using fractionated protein, 30 μg of total protein was separated on Bolt 4–12% Bis-Tris Plus (Thermo Fisher scientific, Waltham, MA) using 100V for 1 hour. The proteins were transferred to a polyvinyl difluoride membrane using the iBlot 2 Dry Blotting system and immune stained by iBind Western Device (Thermo Fisher Scientific, Waltham, MA) using antibodies as follows. The antibodies were Cadherin 11 (#321700, Thermo Fisher scientific), E-Cadherin (#3195), N-Cadherin (#13116), $\beta$-catenin (#9562), GAPDH (#5174), Lamin A/C (#2032), anti-rabbit IgG (#7074,) and anti-mouse IgG (#7076) from Cell signaling Technology. For detection, Immobilon Western Forte HRP Substrate (Millipore, Burlington, MA) and Odyssey Fc Imaging System (LI-COR Biosciences, Lincoln, NE) were used. Western Blot images were quantified using Image Studio Ver 5.2 (LI-COR Biosciences, Lincoln, NE).

## Histopathological analyses

Cells were plated in a density of $5x10^4$ cells/mL/well into chambers of 8 Chamber Cell Culture Slide (CellTreat, Pepperell, MA) and were cultured for 48 hours or treated with WNT3A medium for 2, 6 and 24 hours or L-Cell medium for 24 hours. Following fixation with 4% paraformaldehyde, permeabilization with 3% TritonX-100 and blocking with 2% BSA in PBS. Slides were incubated with primary antibody of Cadherin 11 (#321700, Thermo Fisher scientific) and $\beta$-catenin (#9562, Cell Signaling Technology, Danvers, MA) overnight. Anti-rabbit IgG Alexa Fluor 488 Conjugate (#4412, Cell Signaling) and Anti-mouse IgG Alexa Fluor 647 Conjugate (#4410, Cell Signaling) were used as secondary antibodies. Cytopainter F-Actin Staining Kit Green (#176753, Abcam, Cambridge, UK) was used to assess for morphological changes. ProLong Gold antifade reagent with DAPI (Thermo Fisher Scientific) was used for nuclear counterstaining. Images of the stained slides were captured, and fluorescent intensity was quantified using a BZ-X710 fluorescent microscope (KEYENCE, Itasca, IL). Cytoplasm and neurtite-like cytoplasmic extensions changes were quantified as previously described [26]. Immunohistochemistry and immunofluorescence using formalin-fixed, paraffin-embedded tissue were performed as previously described [28].

## Animal experiments

Tail vein injections were performed by injecting $5.0 \times 10^6$ cells of TC71 or A4573 with CDH11 knocked out by CRISPR/Cas9 or transfected with an empty px458 vector (#48138, addgene, Watertown, MA) as control suspended in PBS directly into the tail vein of NOD/SCID/IL-2Rγ-null (NSG) mice bred by Colorado University Office of Laboratory Animal Resources. Ten mice were used for each cohort. Animals were euthanized 4 weeks after the injections, or when they exhibited signs and symptoms of pain and suffering, such as hunched posture and reluctance to move. After the euthanasia, the lung metastasis was quantified by ddPCR and the number of liver metastasis was counted. All experiments were approved by the Institutional Animal Care and Use Committee of the University of Colorado, and adhered to the standards

set by the National Academy of Science [44]. Animals were monitored for pain and suffering during the experiment (greater than 20% weight loss, lack of spontaneous activity, lack of eating, labored breathing), and were euthanized when symptoms of pain and suffering were observed, to alleviate suffering. Per approved animal use protocol euthanasia method, animal euthanasia was performed with isofluorane or carbon dioxide, followed by thoracotomy for secondary euthanasia. During the animal experiment, animal health and behavior were monitored daily, and weight measured twice weekly.

## DNA extraction and ddPCR detection

Genomic DNA was extracted from mice right lung tissues using a QIAamp DNA Mini kit (Qiagen, Hilden, Germany). DNA concentration was measured by the NanoDrop 2000 (Thermo Fisher Scientific, Waltham, MA) and 30ng DNA (15ng/well) per each sample was used for ddPCR method. The QX200 ddPCR System (Bio-Rad Technologies, Hercules, CA) was used per the manufacturer's protocol using assays for the cell line-specific *EWSR1-FLI1* breakpoint spanning primers and probe sets [45], and Actb (assay ID: dMmuCNS292036842; Bio-Rad Technologies) or Eif2c1(assay ID:dRnoCPE5166213; Bio-Rad Technologies). Instrument control and analysis were performed using QuantaSoft Software (Bio-Rad Technologies). Results were reported as copies/mL after Poisson distribution of occupied to unoccupied droplets was taken into account by the QuantaSoft software.

## Single cell RNA-seq

Ten million A4573 Ewing sarcoma cells were implanted into the pretibial space of NSG mice, and tumors were collected upon growth. Tumors were single cell dissociated using an enzymatic cocktail of 0.1% DNase and Liberase 400μg/mL (Roche/Sigma-Adrich, St. Louis, Missori, USA). Cells were then selected for viability using a BD FACSAria I cell sorter at the Allergy and Clinical Immunology Flow Cytometry Facility at the Division of Allergy and Clinical Immunology, University of Colorado School of medicine. Cells were then combined with the Chromium bead-attached primer library, emulsification oil, and DNA polymerase on a GemCode Chip. The GEM beads were then be thermocycled, the emulsion broken, and DNA recovered per the 10x Genomics protocol. Barcoded single cell libraries were sequenced on an Illumina NovaSEQ 6000 platform at the Genomics Shared Resources at the University of Colorado Cancer Center.

Cell Ranger (v3.1) [46] was used to process the fastq files to cell and gene count tables using unique molecule identifiers (UMI) aligning to GRCh38 (compiled by 10X as 2020-A). The Seurat (v4.0.4) [47] pipeline was used for downstream quality control and analysis. Cell Ranger-filtered data was further processed by removing genes identified in fewer than 10 cells. Cell barcodes were removed if they had greater than 50,000 UMIs, less than 10,000 UMIs, less than 2,000 genes, or greater than 5% of UMIs from mitochondria. Cell doublets were identified and removed using the scDblFinder R package [48]. The gene counts were normalized using Seurat where counts are divided by each cells total counts and multiplied by 10,000 followed by natural-log transformation. The top 2,000 most variable genes were scaled with S score, G2M score, and percentage mitochondria regressed out. Enrichment of select gene sets by cluster was calculated using the singleseqgset R package (https://github.com/arc85/singleseqgset) [49]. Gene set scores were calculated at the cell-level using Seurat's AddModuleScore function and correlation was calculated with the corrplot R package (https://github.com/taiyun/corrplot) [50].

## Statistical analysis

Statistical analyses were performed using GraphPad Prism version 7.00 (GraphPad Software, La Jolla, CA). Post-amputation survival and metastasis-free survival of animals were tested for

significance by the log-rank test. Statistical differences between mean values were evaluated using the Mann-Whitney U test.

## Supporting information

**S1 File.**
(PDF)

**S1 Raw images.**
(PDF)

## Author Contributions

**Conceptualization:** Masanori Hayashi.

**Data curation:** Ryota Shirai, Naoki Oike, Andrew Goodspeed, Masanori Hayashi.

**Formal analysis:** Ryota Shirai, Deandra Walker, Avery Bodlak, Timothy Porfilio, Naoki Oike, Andrew Goodspeed, Masanori Hayashi.

**Funding acquisition:** Masanori Hayashi.

**Investigation:** Ryota Shirai, Tyler Biebighauser, Deandra Walker, Jillian Oviedo, Sarah Nelson-Taylor, Avery Bodlak, Timothy Porfilio, Naoki Oike.

**Methodology:** Ryota Shirai, Sarah Nelson-Taylor, Masanori Hayashi.

**Project administration:** Masanori Hayashi.

**Resources:** Masanori Hayashi.

**Software:** Andrew Goodspeed.

**Supervision:** Masanori Hayashi.

**Validation:** Ryota Shirai.

**Visualization:** Ryota Shirai, Andrew Goodspeed.

**Writing – original draft:** Ryota Shirai, Masanori Hayashi.

**Writing – review & editing:** Ryota Shirai, Masanori Hayashi.

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
