## [Decision Letter · Decision Letter 0]

13 Nov 2023

PONE-D-23-20515Cadherin-11 contributes to the heterogenous and dynamic Wnt-Wnt-β-catenin pathway activation in Ewing sarcoma.PLOS ONE

Dear Dr. Hayashi,

Thank you for submitting your manuscript to PLOS ONE. After careful consideration, we feel that it has merit but does not fully meet PLOS ONE’s publication criteria as it currently stands. Therefore, we invite you to submit a revised version of the manuscript that addresses the points raised during the review process.

Both reviewers find your study interesting , but have raised a number of relevant points that needs to be addressed point by point. I would be happy to grant extra time should this be needed.

We look forward to receiving your revised manuscript.

Kind regards,

Donald Gullberg, PhD

Academic Editor

PLOS ONE

Journal Requirements:

"This work was supported by National Institutes of Health grant (K12HD068372) (M.H.) https://www.nih.gov/grants-funding, St. Baldrick’s Foundation Scholar Award (M.H.) https://www.stbaldricks.org/for-researchers/,the Hyundai Hope on Wheels Young Investigator Award (M.H.), and Tanabe-Bobrow Foundation award (M.H.) https://hyundaihopeonwheels.org/.  This study was partly supported by the National Institutes of Health P30CA046934 Bioinformatics and Biostatistics Shared Resource.

Additional Editor Comments:

Both reviewers find your study interesting , but have raised a number of relevant points that needs to be addressed point by point. I would be happy to grant extra time should this be needed.

Reviewers' comments:

Reviewer's Responses to Questions

**Comments to the Author**

1. Is the manuscript technically sound, and do the data support the conclusions?

Reviewer #1: No

Reviewer #2: Yes

2. Has the statistical analysis been performed appropriately and rigorously? 

Reviewer #1: Yes

Reviewer #2: No

3. Have the authors made all data underlying the findings in their manuscript fully available?

Reviewer #1: Yes

Reviewer #2: Yes

4. Is the manuscript presented in an intelligible fashion and written in standard English?

Reviewer #1: Yes

Reviewer #2: Yes

5. Review Comments to the Author

Reviewer #1: The study by Shirai et al. aimed to investigate the contribution of CDH11 and b-catenin in Ewing sarcoma. This work could have been interesting; however, it lacks rigor and several points need to be clarified:

1. The English and scientific writing could be much improved.

In addition, the figure legends lack information that is important to understand the figures.

References need to be reformatted (e.g., #39 taking half a page!).

2. In general, blots and immunostaining should be quantified.

3. Fig 1: Good expression of b-catenin is still observed in TC71 cells after knockdown (A). So, how the authors can explain a such dramatic effect in the following experiments (like in B&C)? Especially given that basic b-catenin signaling is very low in these cells.

4. Fig 2: heterogeneity of Wnt/b-catenin activation in Ewing sarcoma is already published (reference #20 cited by the authors), and this figure does not bring any new information and should be removed. Moreover, following the methods described, the conclusions driven from this figure are biased: a) The whole A4573 xenograft has been used for the scRNAseq, so this includes mouse cells that have infiltrated the tumor. b) How Wnt signaling can be enriched when it is activated in only 4.3% of the A4573 population in vivo? c) It was performed on A4573 only, this is not representative.

It would have been more interesting to perform scRNAseq on CDH11 KO/WT cells.

5.Fig 4: IP (C) should be repeated in the other way, i.e., with CDH11 IP. The authors mentioned in the text that CDH11 interacted with "non-phosphorylated" b-catenin, but without showing it.

The immunostaining (B) is not convincing, several cells are negative; CDH11 antibody does not look very good.

6. Fig 5: Same comment as for Fig 4, immunostainings are not convincing (F): A4573 showed no expression of b-catenin in the control; we do not see any differences in nuclear localization of b-catenin between the different conditions in TC71 and A4573 cells (quantification is necessary). In addition, WB did not show any b-catenin translocation to nucleus in A4573 after Wnt stimulation (E) in contrast to what it should.

7. Fig 6: In C, counting cells with neurite outgrowth is a bit fishy (on pictures we can clearly see some in all conditions), it would have been better to quantify cell area.

In the cell migration assay (D), It is tricky to associate CDH11 to b-catenin signaling, since in basal condition this pathway is activated in only few cells (and here the control showed good migration). What is the status of b-catenin signaling during cell migration? The assay should be repeated including stimulation with Wnt.

Another type of cell migration like a wound assay could be considered.

8. Fig 7: Using CRISPR/Cas9, the authors have knock-out CDH11 (so the cells do not express it anymore, as seen in the blot 5C; knock-out is different than knock-down using shRNA, where in the latest some expression can still be observed). So why is there huge discrepancies between KO#1 and 2 in 7A and B (in the two cell type??)? In this context, the results does not look reliable.

To what correspond the control? Has flow cytometry analysis been performed for CDH11 expression at the cell surface?

Reviewer #2: The manuscript examines the role of Cadherin-11 in Wnt/β-Catenin signalling in Ewing sarcoma (ES) cells. The authors suggest that cadherin-11 contributes to Wnt/β-Catenin signalling in ES cells by controlling the expression of β-catenin. Using a xenograft model, the authors suggest that loss of cadherin-11 in ES cells can lead to a less metastatic phenotype. In general, the manuscript is technically sound. Although, some of the claims are supported with the experimental data, some major issues remain to be addressed.

Major comments:

1. The results showing the reduction in the protein levels of β-catenin expression in Cadherin-11 KO cells in interesting. In this context, mRNA levels of β-catenin and Cadherin-11 would help to know if the reduction was due to degradation or at transcription level. Authors could also use proteasomal inhibitors MG-132 to demonstrate the role of Cadherin-11 in the regulation or stabilization of β-catenin expression.

2. The data set in figure 5 could improve with quantification and statistical analysis. Importantly, the nuclear-cytoplasmic ratio of β-catenin in the western blots would help to understand if the reduced nuclear β-catenin in cadherin-11 KO cells are due to reduced total expression of β-catenin or due to changes in its localization.

3. The immunofluorescence staining of Cadherin-11 throughout the article is not very convincing to evaluate the expression level. Protein levels of Cadherin-11 in CRISPR-based Cadherin-11 KO cells would be better than immunofluorescence.

Minor comments:

1. The authors state that β-catenin is predominantly localized in the cell membrane. The fractions used for western blotting are cytoplasmic and not purely membrane fractions. Addition of Wnt ligands in experiments related to 3C would be helpful to know if β-catenin is predominantly localized in the cytoplasmic fractions despite activation of Wnt/β-Catenin signalling. Based on figure 2A, it seems there is very little activation of Wnt/β-Catenin signalling in the absence of Wnt ligands and this could explain the reduced nuclear localization of β-Catenin

2. For immunofluorescence the combination of 488 and 647 would be better instead of 488 and 555 to avoid cross-excitation issues mainly for cadherin-11 staining.

3. Figure legends are not completely clear enough to understand the data. For example, it is not clear what is Lcell and RPMI. Also, in figure 5C its mentioned as CDH11 KO but mentioned as CDH11 KD in 5E.

6. PLOS authors have the option to publish the peer review history of their article (what does this mean?). If published, this will include your full peer review and any attached files.

Reviewer #1: No

Reviewer #2: No

---

## [Author Response · Author response to Decision Letter 0]

23 Apr 2024

We would like to thank the two reviewers for a thorough review and the valuable input from the initial review. Below, we have attached a point to point answer to the reviewer comments. With these revisions, we believe the manuscript has become significantly stronger. We hope you will agree that this report is now suitable for publication in PLOS ONE. 

Sincerely, 

Masanori Hayashi, MD

Editor comments: 

Format has been corrected.

This has now been added to the methods section, “Animal Experiments”. 

The included single cell RNA-seq data will be deposited to GEO. 

"This work was supported by National Institutes of Health grant (K12HD068372) (M.H.) https://www.nih.gov/grants-funding, St. Baldrick’s Foundation Scholar Award (M.H.) https://www.stbaldricks.org/for-researchers/,the Hyundai Hope on Wheels Young Investigator Award (M.H.), and Tanabe-Bobrow Foundation award (M.H.) https://hyundaihopeonwheels.org/. This study was partly supported by the National Institutes of Health P30CA046934 Bioinformatics and Biostatistics Shared Resource.

We have included the amended funding statement within our new cover letter. 

Data Availability has been updated in the cover letter and data availability statement to specify that the minimal data set has been submitted. All blots are now uploaded as S1 Raw, and genomic data has been uploaded to GEO.

This has been clarified. Raw blot data has been submitted as a supplemental file. 

We included a statement that the animal studies were approved by our local IACUC. No human samples were used, so no disclosure for IRB. 

Additional Editor Comments:

Both reviewers find your study interesting , but have raised a number of relevant points that needs to be addressed point by point. I would be happy to grant extra time should this be needed.

Reviewer comments:

Reviewer #1: The study by Shirai et al. aimed to investigate the contribution of

CDH11 and b-catenin in Ewing sarcoma. This work could have been interesting; however, it lacks rigor and several points need to be clarified:

1. The English and scientific writing could be much improved.

In addition, the figure legends lack information that is important to understand the figures.

References need to be reformatted (e.g., #39 taking half a page!).

The main document was proofread by a native English speaker and all sections have been significantly modified. Figure legends were revised with further details. In our original submission, we had used Endnote to format the references according to the PLOS template. For reference 39, we found that Endnote will automatically format this reference in this fashion due to a formatting requirement that appears to force all the COI disclosures to be included. We could not find any other way to format this reference, so we removed this and replaced it with a similar report that can still emphasize the same point.

2. In general, blots and immunostaining should be quantified.

All western blots were quantified using Image Studio Ver5.2 from LI-COR and immunostaining images were quantified using BZ-X710 from Keyence.

3. Fig 1: Good expression of b-catenin is still observed in TC71 cells after knockdown (A). So, how the authors can explain a such dramatic effect in the following experiments (like in B&C)? Especially given that basic b-catenin signaling is very low in these cells.

This is an excellent point, considering the relatively heterogeneous expression of �-Catenin in these cells. We do not have an exact explanation on why colony formation and soft agar formation is so dramatically decreased. However, to maintain scientific objectivity, we also never make any claim that the degree of decrease in Fig 1B-C is nothing more than statistically significant. We speculate that even the degree of knockdown accomplished in Fig 1A is enough to significantly decrease the colony formation and soft agar formation, because a very small number of cells within the general population is Wnt/�-Catenin active at a time (as shown in Fig 2), and these cells are critical to the colony formation. 

4. Fig 2: heterogeneity of Wnt/b-catenin activation in Ewing sarcoma is already published (reference #20 cited by the authors), and this figure does not bring any new information and should be removed. Moreover, following the methods described, the conclusions driven from this figure are biased: a) The whole A4573 xenograft has been used for the scRNAseq, so this includes mouse cells that have infiltrated the tumor. b) How Wnt signaling can be enriched when it is activated in only 4.3% of the A4573 population in vivo? c) It was performed on A4573 only, this is not representative.

It would have been more interesting to perform scRNAseq on CDH11 KO/WT cells.

In the original Pedersen work that we cited, Wnt/�-Catenin activation heterogeneity was only demonstrated in cell cultures, and in vivo analysis was limited to IHC of �-Catenin in archival tumors. We included this figure due to two reasons. First, our work contains mostly in vivo results in a mouse model, demonstrating heterogeneity of Wnt/�-Catenin signaling, a finding that Pedersen did not demonstrate. Second, even if one paper has published a similar figure in the past using a different cell line, we feel that demonstrating that we can reproduce some of Pedersen’s conclusions (Fig2A is the only figure that is close to Pedersen) is important for scientific rigor. To avoid confusion, we have emphasized in the new version that we are merely corroborating published data for Fig2A, and that the new addition to literature is the findings in vivo. In figure2B, we demonstrate that 4.3% of tumor cells were activated (representative figure), indicating a small subpopulation of activated cells exist in vivo. We have done this experiment in triplicate, and the data is now demonstrated as a comparison to baseline activation in vitro that was observed in Fig 1A. This is why we conclude there is heterogeneity in Wnt activation within the tumor. In figure B-E, our original intention was to possibly do this in two cell lines. However, the single RNA-seq experiment for the TC71 derived xenograft did not pass our QC. Violin plot shown below. Considering the significant cost of single RNA-seq, and the fact that this figure is not the main point of the paper, we chose to only include A4573 xenograft data. Considering the criticism of only having one cell line, we have significantly toned down the discussion and results. 

As standard QC, we remove all cells from the xenograft scRNA-seq data that aligned to the mouse transcriptome, and the data shown are from all human cells that have passed QC. 

5.Fig 4: IP (C) should be repeated in the other way, i.e., with CDH11 IP. The authors mentioned in the text that CDH11 interacted with "non-phosphorylated" b-catenin, but without showing it.

The immunostaining (B) is not convincing, several cells are negative; CDH11 antibody does not look very good.

We have attempted CDH11 IP, however this was technically difficult due to a lack of a robust enough antibody. We have removed the text mentioning non-phosphorylated �-catenin. The immunostaining was revised according to comments from reviewer 2 using different secondary antibodies. In our new experiment, we used an Alexa Fluor 488 conjugate secondary antibody for �-Catenin, and Alexa Fluor 647 conjugate secondary antibody for CDH11, and the immunostain is much more robust.

6. Fig 5: Same comment as for Fig 4, immunostainings are not convincing (F): A4573 showed no expression of b-catenin in the control; we do not see any differences in nuclear localization of b-catenin between the different conditions in TC71 and A4573 cells (quantification is necessary). In addition, WB did not show any b-catenin translocation to nucleus in A4573 after Wnt stimulation (E) in contrast to what it should.

We have removed the immunostains included in the previous Fig 5 B D and F. As other figures in the paper, we have now quantified the western blot imaging. For the previous Figure 5E (now in supplemental), the sensitivity of western blots in detecting protein from a small subpopulation is so poor that we feel it did not prove nor disprove the hypothesis. Therefore, we moved this figure to supplemental, and have noted such in the main text. For the previous Fig 5F (now Fig 5C), we have added statistical analysis. 

7. Fig 6: In C, counting cells with neurite outgrowth is a bit fishy (on pictures we can clearly see some in all conditions), it would have been better to quantify cell area.

In the cell migration assay (D), It is tricky to associate CDH11 to b-catenin signaling, since in basal condition this pathway is activated in only few cells (and here the control showed good migration). What is the status of b-catenin signaling during cell migration? The assay should be repeated including stimulation with Wnt.

Another type of cell migration like a wound assay could be considered.

In Fig 6C, we have now quantified the cell area, and performed statistical analysis as recommended. We agree that claiming that Fig 6 proves that CDH11 knockout leading to decreased �-Catenin signaling activation is an over-representation of the data, and we have amended the text to tone it down. We have attempted Wnt stimulation experiments for cell migration, but we have faced significant technical difficulties due to the background of the system we are using. We have been performing Wnt ligand stimulation using a Wnt3a secreting L-cell system. In this system, the Wnt3a media (or control L-Cell media), is collected as a supernatant from these cells after incubation. This supernatant is often nutrient poor (since it has been used by L-cells for 48-96 hours), DMEM based (All of our Ewing sarcoma cells are cultured in RPMI), and Ewing sarcoma cell lines do not grow normally in these media. Which is why we have limited stimulation experiments to Fig2A and Fig 5, where we only quantify the �-catenin signaling subpopulation after Wnt stimulation. When we used this nutrient poor DMEM based supernatant and tried to perform migration experiments, the cells did not migrate enough even at baseline to have any measurable statistical change. Because of this limitation, we were unsuccessful in repeating the assay with Wnt stimulation.

8. Fig 7: Using CRISPR/Cas9, the authors have knock-out CDH11 (so the cells do not express it anymore, as seen in the blot 5C; knock-out is different than knock-down using shRNA, where in the latest some expression can still be observed). So why is there huge discrepancies between KO#1 and 2 in 7A and B (in the two cell type??)? In this context, the results does not look reliable.

To what correspond the control? Has flow cytometry analysis been performed for CDH11 expression at the cell surface?

We have performed flow cytometry to demonstrate CDH11 cell surface expression was depleted in knockout conditions, in Fig 5, where we initially demonstrate successful knockout. The conclusion of Fig 7A and 7B has been clarified, since they are distinct ex

---

## [Decision Letter · Decision Letter 1]

31 May 2024

Cadherin-11 contributes to the heterogenous and dynamic Wnt-Wnt-β-catenin pathway activation in Ewing sarcoma.

PONE-D-23-20515R1

Dear Dr. Hayashi,

"We have had problems to reach one of the original reviewers, hence the delay.  Instead one editor has looked at your response to the non-responding reviewer and together with the positive response from the second reviewer we are glad to accept your revised version.

Donald Gullberg"

We’re pleased to inform you that your manuscript has been judged scientifically suitable for publication and will be formally accepted for publication once it meets all outstanding technical requirements.

Kind regards,

Donald Gullberg, PhD

Academic Editor

PLOS ONE

Additional Editor Comments (optional):

Reviewers' comments:

Reviewer's Responses to Questions

**Comments to the Author**

1. If the authors have adequately addressed your comments raised in a previous round of review and you feel that this manuscript is now acceptable for publication, you may indicate that here to bypass the “Comments to the Author” section, enter your conflict of interest statement in the “Confidential to Editor” section, and submit your "Accept" recommendation.

Reviewer #1: All comments have been addressed

2. Is the manuscript technically sound, and do the data support the conclusions?

Reviewer #1: (No Response)

3. Has the statistical analysis been performed appropriately and rigorously? 

Reviewer #1: (No Response)

4. Have the authors made all data underlying the findings in their manuscript fully available?

Reviewer #1: (No Response)

5. Is the manuscript presented in an intelligible fashion and written in standard English?

Reviewer #1: (No Response)

6. Review Comments to the Author

Reviewer #1: (No Response)

7. PLOS authors have the option to publish the peer review history of their article (what does this mean?). If published, this will include your full peer review and any attached files.

Reviewer #1: No

---

## [Editor Report · Acceptance letter]

5 Jun 2024

PONE-D-23-20515R1 

PLOS ONE

Dear Dr. Hayashi, 

I'm pleased to inform you that your manuscript has been deemed suitable for publication in PLOS ONE. Congratulations! Your manuscript is now being handed over to our production team.

Kind regards, 

on behalf of

Professor Donald Gullberg 

Academic Editor

PLOS ONE